# Exploring Zika Virus Impact on Endothelial Permeability: Insights into Transcytosis Mechanisms and Vascular Leakage

**DOI:** 10.3390/v16040629

**Published:** 2024-04-18

**Authors:** Dama Faniriantsoa Henrio Marcellin, Jufang Huang

**Affiliations:** Department of Anatomy and Neurobiology, School of Basic Medical Sciences, Central South University, Changsha 410013, China; damafaniriantsoahenrio@gmail.com

**Keywords:** Zika virus (ZIKV), blood–brain barrier (BBB), transcytosis, endothelial cells

## Abstract

Treating brain disease is challenging, and the Zika virus (ZIKV) presents a unique obstacle due to its neuroinvasive nature. In this review, we discuss the immunopathogenesis of ZIKV and explore how the virus interacts with the body’s immune responses and the role of the protein Mfsd2a in maintaining the integrity of the blood–brain barrier (BBB) during ZIKV neuroinvasion. ZIKV has emerged as a significant public health concern due to its association with severe neurological problems, including microcephaly and Gillain–Barré Syndrome (GBS). Understanding its journey through the brain—particularly its interaction with the placenta and BBB—is crucial. The placenta, which is designed to protect the fetus, becomes a pathway for ZIKV when infected. The BBB is composed of brain endothelial cells, acts as a second barrier, and protects the fetal brain. However, ZIKV finds ways to disrupt these barriers, leading to potential damage. This study explores the mechanisms by which ZIKV enters the CNS and highlights the role of transcytosis, which allows the virus to move through the cells without significantly disrupting the BBB. Although the exact mechanisms of transcytosis are unclear, research suggests that ZIKV may utilize this pathway.

## 1. Introduction

One of the diseases that is poorly understood and treated is brain disease. Despite the recent advances in drug discovery, effective treatments for the central nervous system (CNS) [1], and understanding of neuroinvasion, the success rate remains low. Reaching macromolecular targets in the brain is extremely difficult due to the brain tissue’s carefully regulated extracellular environment, which is a significant obstacle to therapeutic efficacy. Some barriers protect it from the bloodstream, including the blood–brain barrier (BBB) and the blood–cerebrospinal fluid barrier. This makes the CNS an immune-privileged organ [2].

Zika virus (ZIKV) is a flavivirus that is transmitted by the bite of an infected mosquito of the genus *Aedes* called *Ae. Aegypti* and *Ae. Albopictus*. It has a single-stranded, positive-polarity RNA genome, which includes seven non-structural (NS) proteins, NS1, NS2A, NS2B, NS3, NS4A, and NS5; and three structural proteins, capsid (C), pre-membrane/membrane (prM/M), and envelope (E) [3]. Various bodily fluids, such as saliva, tears, urine, and semen, as well as in the brain, female vaginal tract, and testis, have all been described to contain ZIKV [4]. ZIKV infections are typically asymptomatic [5]. Still, symptoms like rash, fever, and headaches can occur after 3–14 days of incubation [6], potentially leading to severe neurological sequelae, including microcephaly, meningoencephalitis, myelitis, and Guillain–Barré syndrome (GBS) [7]. In the pregnant woman, ZIKV infection is notably associated with microcephaly, which is a condition where the head circumference is less than three standard deviations (SDs) below the mean and is influenced by gender and gestational age [8,9]. Brain abnormalities associated with ZIKV can manifest without microcephaly, making neuroimaging a critical component of prenatal diagnosis [9,10]. Similarly, GBS is associated with ZIKV infection, characterized by acute flaccid paralysis and an autoimmune polyneuropathy following infection in adults, which is caused by an overreactive immune response [7,11]. Brain development during pregnancy is disturbed, leading to cranial collapse and disruption of neuronal and glial migration, which are the causes of ZIKV-associated brain abnormalities [12]. In addition, ZIKV is also found to be associated with Congenital Zika Syndrome (CZS) and long-term developmental consequences for affected children [8]. The World Health Organization (WHO) and the Pan American Health Organization (PAHO) have published clinical diagnostic standards for ZIKV [13,14]. However, the precise mechanism of ZIKV-induced neurological damage remains poorly understood, and further research in this area is needed.

Together with the other members of the Flaviviridae family—West Nile virus (WNV), dengue virus (DENV), hepatitis C virus (HCV), and Japanese encephalitis virus (JEV)—ZIKV has been classified as neurotropic and has significant neuroinvasive properties. Due to its association with more severe neurological symptoms, ZIKV has recently attracted worldwide attention [15]. Recent research has emphasized its neuroinvasive nature, affinity, and potential damage. Neonatal ZIKV infection causes aberrant vascular density and diameter in the developing brain, resulting in a leaky BBB and significant neuronal loss [16]. Given the severe impacts of these complications, a detailed understanding of ZIKV’s pathogenesis underlying mechanisms, especially its effect on brain development and BBB, is crucial for developing effective therapies [17]. Despite these severe side effects, knowledge of ZIKV neuroinvasion and pathogenesis in the developing brain remains elusive.

Before ZIKV can invade the brain of the infected mother and infect the fetus, it must overcome two obstacles, namely the placental barrier (BPB) [18] and the BBB [19]. However, the mechanism underlying this infiltration is not yet fully understood. The placenta, a highly specialized organ that develops only during pregnancy, supports the growth and development of the fetus [1]. Its primary purpose is to supply the developing embryonic brain with enough oxygen and nutrients. However, the ZIKV has taken this as an opportunity to invade the CNS. Moreover, the placental barrier acts as a vital physiological barrier that protects the fetus from harmful chemicals, maternal diseases, and pathogenic infections such as ZIKV and DENV [20]. Instead, it can serve as a conduit for the spread of the virus from the infected mother to the fetus. The placenta, while usually shielding most pathogens from reaching the fetus’s brain, can also facilitate the transmission of the ZIKV from an infected mother to her fetus. In cases where the placenta is infected with ZIKV, it can lead to chronic placentitis [21], which specifically affects macrophages [22] and trophoblast cells [23].

The placenta acts as the first barrier for the fetus, protecting it from pathogens and promoting its growth and development. The BBB acts as the fetus’ second barrier, protecting the fetal brain and ensuring healthy brain development [19]. It is an intricate structure composed of tightly connected brain microvascular endothelial cells (BMECs) associated with pericytes, astrocytes, and microglia. Between the blood and the CNS parenchyma, this structure serves as a barrier that prevents the entry of pathogens, such as viruses or virus-infected cells [24,25]. The cells associated with the BBB have a variety of pattern recognition receptors (PRRs) that enable them to respond to damage-associated molecular patterns (DAMPs) generated by CNS injury and to pathogen-associated molecular patterns (PAMPs) acquired by viral infections. The barrier function can, therefore, be impaired by an inflammatory insult or other pathobiological conditions that affect their metabolism [26].

BMECs are unfenestrated polarized endothelial cells connected by continuous tight junctions (TJs) [3]. The junctions consist of TJ proteins called claudins, occludins, and adhesion molecules, which allow cells to stick together: E-cadherins and VE-cadherins. The TJ complexes are stabilized by interacting with the intracellular cytoskeleton via adaptor proteins such as zonula occludin (ZO) and others [27]. More than 40 proteins are found in the TJ, and they act as gatekeepers for the paracellular route and prevent the mixing membrane proteins between the apical and basolateral membranes (Figure 1). Some viruses and parasites utilize BMECs for replication, potentially damaging BMEC and releasing progeny [28]. Once TJ and BBB are disrupted, an infection of CNS appears, increasing endothelial cell permeability and allowing pathogens and toxins to reach the brain [29]. Neurotropic viruses have been shown to enter the CNS via multiple routes [3]. For instance, JEV, WNV, ZIKV, and TBEV can cross the endothelial barrier as cell-free viruses and infected peripheral immune cells, such as monocytes and leukocytes, via hematogenous routes to enter the CNS [19], which is the most frequent neuroinvasion pathway for flaviviruses [27]. Neurotropic viruses and fungi can utilize caveolae-dependent or -independent endocytosis [30]. Leukocytes infected with pathogens can also cross the BBB and release them into the perivascular space due to the indirect effects of systemic inflammatory cytokines such as tumor necrosis factor (TNF) and IFN or direct binding to claudins [31]. Pathogens can disrupt the BBB by damaging the BMECs or degrading TJs [32]. This TJ degradation is caused by proteases from both the pathogens and the host that are triggered by inflammation. Bacterial toxins can also damage BMECs, leading to BBB permeability and injury of the CNS [4,28,32].

On the other hand, pathogens can cross the BBB transcellularly while maintaining barrier integrity [24]. ZIKV was released basolaterally in a transwell assay without inducing a significant cytopathic effect in the BBB cells [33]. Similarly, a recent study showed that infection of HBMECs with African and Brazilian ZIKV strains (ZIKVMR766 and ZIKVPE243) resulted in effective viral replication and basolateral release of infectious particles that were able to replicate in other cell types seeded separately in a transwell system [24]. The maintenance of TJ protein expression and localization showed that the viruses were able to pass through the endothelial barrier without increasing permeability [24]. However, the transcellular process involving transport within endothelial cells by transcytosis during ZIKV infection remains unclear. Despite the intact permeability of the monolayer following the ZIKV infection of HBMECs, uncertainties remain regarding the specific transcytosis mechanisms utilized by ZIKV to enter the CNS [34]. Furthermore, it is necessary to understand the ability of ZIKV to affect BBB cells (Figure 1).

In this review, the relationship between the BBB, placental barrier, and endothelial cells forms crucial defense mechanisms against pathogens like ZIKV. Despite our growing understanding of the molecular mechanisms involved, the answers to many questions regarding ZIKV neuroinvasion and its impact on endothelial permeability remain unknown. We provide an overview of the possible strategies that ZIKV has been shown to use to penetrate the BBB and the processes by which it can infiltrate the CNS and cause inflammation in the brain. We also discuss the physiopathology of vascular permeability and ZIKV-associated neurological and cognitive impairments. Through these explorations, we aim to shed light on the multifaceted nature of ZIKV infection and its implications for neurological sequelae. This work can also help researchers and scientists elucidate insights into current vaccine and drug development trends, aiding in the quest for effective strategies against ZIKV infection and answering critical questions that remain unanswered.

## 2. Immunopathogenesis of ZIKV

To better understand the complex ZIKV disease, diagnosis, prevention, and treatment, it is essential to recognize the importance of specific ZIKV lineage; the impact of comorbidities; the viral load; and the molecular mechanisms underlying severe disease, such as genetic susceptibility of the host to infection, immunosuppression, and failure of innate immunity [35,36,37,38]. In general, two phases can be distinguished in the host immune response to ZIKV that are similar to those of other flaviviruses. Firstly, the innate immune response consists of the initial phase of pathogen identification and the onset of a general non-specific antiviral defense, followed by an adaptive immune response, which is crucial for establishing and developing an effective defense against infection [39,40]. Generally, ZIKV’s innate immunity is activated when the host cell responds to the viral RNA PAMPs through the retinoic acid-inducible gene I (RIG-I), one of the RIG-I-like receptors (RLRs) [36,41]. As a result of this process, the innate immune system genes are expressed, which also activates latent transcription factors such as NF-κB and interferon regulatory factor 3 (IRF3) [42]. Once IRF3 is activated, it can lead to the induction of type I and III interferons (IFN) via the Jak-STAT pathway, and then it promotes the transcription of approximately 100 IFN-stimulated genes that mediate immune modulation and antiviral defense [43,44]. The antiviral effect of the innate immune system gene limits the multiplication and spread of ZIKV [43]. A recent study has shown that ZIKV can inhibit Jak-STAT signaling to evade the innate immune defenses of host cells [44]. Significantly, the inadequate programming of the innate immune system may promote a disease state in which ZIKV has fewer barriers to spread and replication. The most pathogenic flavivirus strains have a greater tendency to antagonize the innate immunity of the host compared to avirulent isolates [39,45]. Recent research has shown that ZIKV isolates the same characteristics, with differential activation of the innate immune system and antagonistic interactions between African and Asian lineages [36,46,47,48].

### 2.1. ZIKV Receptor and Its Immune Response

ZIKV entry is facilitated by AXL, a type of receptor tyrosine kinase [49,50], and has been associated with many cellular reactions and the control of inflammatory responses [51]. Once the virus has invaded many cells, tissues, and organs during early infection, the interaction between ZIKV and immune responses begins. Interestingly, the virus quickly leaves the blood of infected humans and non-human primates. However, it can persist for months in saliva, urine, semen, breast milk, and the CNS [52,53,54]. ZIKV is internalized by a mechanism known as clathrin-mediated endocytosis after being attracted to the AXL receptor by the TAM ligand growth arrest-specific 6 (Gas6) [55]. The ZIKV-containing vesicles are then transported to Rab5+ endosomes. AXL kinase activity is activated by the ZIKV/Gas6 complex, which in turn induces transcription of many interferon-stimulated genes (ISGs), TOLL-like receptor 3 (TLR3), DExD/H-box helicase 58 (DDX58), and interferon-induced helicase C domain 1 (IFIH1) [55]. This activation suppresses the innate immune response, facilitating productive infection [55,56]. According to several in vitro and ex vivo studies, ZIKV is known to replicate in a variety of human cells, including endothelial and epithelial cells [57], peripheral blood mononuclear cells (PBMCs) [57], astrocyte and microglial cells [55], trophoblasts [58], Hofbauer cells in chorionic villi and amniotic epithelial cells [58,59], and fibroblasts [60] (placental, uterine, and pulmonary) [59] (Figure 2). Concurrently, these particular cell type's infections are linked to higher AXL protein levels [50]. For instance, in vitro has shown that the infection of glial cells by ZIKV is inhibited by the inhibition of AXL [61]. Similarly, It has also been shown that pretreatment of astrocytes with antibodies directed against AXL reduces ZIKV infection [62]. On the other hand, microglial cells from mice expressing less AXL were found to be immune to ZIKV [55]. Nevertheless, the data suggest that AXL may not be necessary for the ZIKV to enter the body, and its significance varies depending on the kind of cell it infects. When comparing AXL knockout mice to wild-type mice, studies found comparable amounts of ZIKV RNA [62]. It was also found that neural progenitor cells (NPCs) can be infected with ZIKV even without AXL [63]. It is, therefore, possible that additional receptors help the ZIKV to enter specific cells.

### 2.2. Endothelial Immune Response

ZIKV has been found to specifically target NPCs, which are an essential component of the developing embryonic brain. According to some scientists [16,66,67], the primary etiology of neurological sequelae, including microcephaly, is an NPC differentiation disorder, leading to brain injury [67]. Studies comparing different ZIKV strains have revealed variations in infection rates and gene expression profiles in human NPCs. For instance, the Asian strain (FSS13025) resulted in only 46.7% of infections at a higher MOI (0.04) after 64 h of infection, while the African ZIKV strain (MR766) indicated a more significant proportion of infections rate of 69.8% at a lower MOI (0.02) [68].

The responses of endothelial cells to inflammatory stimuli can be distinguished into two types: type I is triggered by thrombin, whereas inflammatory cytokines induce type II. The release of pre-synthesized molecules is the basis for type I response and does not require new gene expression. This process allows the highly rapid activation of the white blood cells, such as leukocytes, to be recruited within minutes of the initial shock [69]. Usually, this response is mediated by G protein-coupled receptors, and their transitory signaling guarantees that the response is transient [70]. The free Ca^2+^ level in the cytosol increases due to G protein-coupled signaling and involves the exocytosis of the Weibel–Palade bodies. In addition, this response is accompanied by the pro-thrombotic protein von Willebrand factor (VWF), P-selectin [71], angiopoietin 2 (ANGPT2) [72], and the potent neutrophil chemoattractant CXCL8 [73]. ZIKV infects endothelial cells and causes vascular leakage and leukocyte recruitment, promoting increased blood flow and vascular leakage due to increased intracellular Ca^2+^ and prostacyclin synthesis, leading to CNS injury.

On the other hand, NF-kB and activation protein 1 (AP-1) are two pro-inflammatory transcription factors, and de novo gene expression is activated by type II responses. Generally, this is the traditional response to pro-inflammatory cytokines such as IL-1 and TNF. Similar studies have demonstrated that ZIKV infects BBB cells highly efficiently, leading to the upregulation of inflammatory cytokines IL-6, IL-8, and chemokines (CCL5 and CXCL10) in vitro and in vivo [25]. These inflammatory molecules not only modulate the integrity of the BBB but also recruit immune cells. In addition, ZIKV infection led to an increase in the expression of cell adhesion molecules (CAMs), which play a role in the attachment of leukocytes to the BBB and are essential for the infiltration of immune cells into the CNS, leading to neuroinflammation [25]. A reduction in the expression of genes associated with extracellular matrix (ECM) organization and collagen synthesis, including collagen-encoding genes that are critical for brain and BBB development, was observed in postmortem brain samples from neonates infected with ZIKV that developed CZS. There was a notable increase in protein tyrosine phosphatase receptor type Z1 (PTPRZ1), an enzyme that regulates inflammation in the CNS [74]. Specific adhesion and inflammatory molecules were more upregulated in the African ZIKV strain than in the Asian strain [25]. Gurung et al. conducted a study in which fetal damage observed in primates infected with ZIKV in pregnancy included pathology in immature oligodendrocytes, dysfunctional migration of neurons into cortical layers, and defects in radial glia. Elevated IL-6 levels, astrogliosis, and microglia were identified as indicators of severe neuroinflammation in the neonates [75]. Neuroinflammation and cortical atrophy in the postnatal brains of rodents have been observed as a result of intrauterine ZIKV infection during pregnancy [76]. Significantly, fetal neurodevelopmental defects were noted despite the absence of detectable ZIKV RNA, suggesting that neuroinflammation may play a role in the development of long-lasting complications [75]. These pro-inflammatory activations lead to a prolonged and more effective inflammatory response, increased blood flow, vascular leakage, and coordinated recruitment of leukocytes.

Overall, activation of both type I and type II endothelial cells leads to increased vascular permeability, which is thought to be due in part to the stimulation of fibrillar adhesions to the extracellular matrix that destabilizes endothelial adherens junctions and allows for the extravasation of plasma proteins [77]. One study found that introducing ZIKV (MEX1-44) into NPCs derived from developing mouse brains resulted in a significant upregulation of TNF-α [16]. Neuronal survival and neurogenesis have been linked to the pro-inflammatory cytokine TNF-α, which accomplishes this by stimulating two receptor subtypes, TNF-R1 and TNF-R2. Other studies have shown that TNF-R1 inhibits the proliferation of progenitor cells, while TNF-R2 promotes the survival of newly formed neurons [78]. Kim et al. [79] found that administration of TNF-α to primary human NPCs inhibited apoptosis via activation of the NF-κB signaling pathway. Additionally, previous studies have shown that TNF-α suppresses the expression of specific neuronal cytoskeletal proteins and the number of neuronal cells [80]. Likewise, TNF-α has been shown to inhibit neuronal differentiation of human NPCs via activation of STAT3 signaling [81]. The overall effect of TNF-α on NPCs is likely influenced by several factors, including cytokine concentration, the affinity and relative expression of TNF-R1 and TNF-R2, the binding to TNF receptors, and subsequent intracellular signaling [82].

Furthermore, the molecular mechanisms underlying disease or protection can be better understood by comparing multiple ZIKV isolates, including the role of sequence variations in virulence, tissue tropism, pathology, and immune evasion. In this regard, specific CD4+ T cells and neutralizing antibodies have been associated with the protective response observed following infection with the ZIKVPE243 strain [83], suggesting that a systemic response may influence the amount of virus reaching the BBB and the extent of the lesion, ultimately impacting disease progression. In addition, Jurado and colleagues investigated the effects of IFNAR deficiency in hematopoietic and non-hematopoietic cells in a mouse model of ZIKV infection. The IFN responses of non-hematopoietic cells were shown to be critical in preventing viral spread to the brain, BBB breakdown, and ZIKV-induced neuropathology. The dependence of virus-induced paralysis on CD8+ T-cell infiltration in the brain suggests that leukocyte infiltration caused by BBB disruption may be a significant problem for neuropathogenesis. In the brains of mice at risk, they also found that astrocytes were particularly affected [84]. These results suggest that astrocyte infection may significantly affect subsequent BBB degradation and lymphocyte infiltration following viral passage via BMECs.

## 3. ZIKV Pathophysiology of Vascular Permeability

ZIKV infection is associated with increased vascular permeability, similar to other flaviviruses [85]. The pathophysiology of this vascular permeability is based on the ability of the virus to disrupt the endothelial barrier, leading to the leakage of fluids and proteins into the surrounding tissue [85,86]. In this regard, the increased vascular permeability manifests in three distinct forms within the BBB that are due to pathologic changes: baseline vascular permeability (BVP), acute vascular hyperpermeability (AVH), and chronic vascular hyperpermeability (CVH).

AVH develops in response to exposure to various vascular permeabilizing factors, such as histamine, vascular endothelial growth factor (VEGF), and tumor necrosis factor-α (TNF-α), which are associated with inflammation and infection [87]. Following this exposure response, two distinct mechanisms occur, namely the destabilization of adherens junctions (AJ) and the activation of actomyosin contractility in endothelial cells, which include c-Src, Ca^2+^ signaling pathways, and RhoGTPase control. Phosphorylated VE-cadherin and β-catenin are internalized and disassembled, leading to the destabilization of the AJ, while the phosphorylation of myosin light-chain kinase (MLCK) triggers actomyosin contraction [88]. These mechanisms dynamically regulate the opening and closing of endothelial cell–cell junctions, leading to the reversible contraction of endothelial cells and the dissociation and reassembly of junctional complex proteins [89]. Consequently, the influx of fluids, solutes, and pathogens into the tissue and extravascular spaces occurs via the paracellular pathway due to the disruption of interendothelial junctions [89].

In conditions of increased permeability like CVH, transient stimuli lead to brief and reversible leakage, leading to acute/chronic inflammation. Significantly, it is mediated by vascular permeabilizing agents such as vascular permeability factor (VPF), VEGF, and VEGF-A, which promote plasma extravasation and increase the permeability of the vascular [86,90]. Leukocyte adhesion and migration can be increased by VEGF-A, which also stimulates endothelial cell sprouting and vasodilation. However, the mechanisms behind these effects probably differ from those controlling endothelial barrier function [86,91]. It can also be sustained in chronic inflammation where abnormal conditions persist, and the microvasculature undergoes remodeling into a more leaky phenotype [92]. Recent research indicated that flavivirus non-structural protein 1 (NS1) increases endothelial cell monolayer permeability in vitro and causes vascular leakage in vivo, both of which are essential steps in pathogenesis [93]. In addition, the endothelial glycocalyx, a network of glycosaminoglycans (GAGs), proteoglycans, and sialic acid (Sia) expressed on the surface of human endothelial cells, is partially responsible for this hyperpermeability [94]. More significantly, elevated levels of these GAGs in the bloodstream are linked to severe cases, especially to other flaviviruses, such as DENV [94]. However, the impact of ZIKV NS1 on the placental barrier’s integrity has not been discussed.

Conversely, the vascular system is crucial for supplying nutrients to human tissues and removing metabolic waste from tissues. The endothelium of the vascular system, which lines the inner layer of the entire circulatory system, consists of a single layer of endothelial cells. It controls the movement of chemicals and fluids between the circulatory system and the tissue compartments of the body by acting as a semipermeable barrier (Figure 1) [95,96]. To support the transport of proteins, solutes, and fluid and to protect the brain, endothelial cells can control their paracellular and transcellular pathways [95]. However, ZIKV is susceptible in BBB endothelial cells, thus giving ZIKV access to cell types such as astrocytes, microglia, and neural stem cells to infect the brain. It also serves as a reservoir that promotes ZIKV’s spread throughout the BBB [39]. The BBB endothelial cells have the potential to release matrix metalloproteinase 2 (MMP-2), which is responsible for disrupting impermeability and making it easier for effector immune cells to enter the brain [97]. By affecting the intercellular connections that bind the individual brain cells and make up the external BBB, ZIKV can lead to a breakdown in the vascular impermeability function [98]. Brain cells infected with ZIKV can release MMP-2, destroying intercellular connection proteins and increasing vascular permeability [92]. Previous studies have shown that ZIKV can infect primary human brain endothelial cells and release the virus from the basolateral side, indicating a potential pathway for viral entry into the brain [99,100]. The comprehensive in vivo models have not yet elucidated the exact mechanism by which ZIKV infiltrates the brain. However, an in vitro model using differentiated brain endothelial cells has shown that ZIKV can cross the BBB with or without compromising its structural integrity [101]. Infection of astrocytes, which are closely associated with endothelial cells, is involved in BBB disruption [100] and provides a route for viral access to the brain. Nevertheless, the initial infection of astrocytes by ZIKV and the mechanisms underlying viral entry into the endothelial layers of the brain remain unclear [100].

Recent research has also suggested that ZIKV attacks pericytes in the choroid plexus and meninges before spreading to the cerebral cortex [102]. Besides, ZIKV can infect human brain vascular pericytes in vitro, resulting in increased endothelial barrier permeability. This regard suggests that infection of pericytes may facilitate ZIKV entry into the CNS [101]. In addition, ZIKV has been observed to infect human retinal pericytes and endothelial cells of the inner BRB in vitro [103]. This infection is associated with triggering increased production of pro-inflammatory cytokines, chemokines, and angiogenic factors, possibly contributing to the development of congenital eye disease commonly observed in microcephalic infants following ZIKV infection [103]. These inflammatory mediators can disrupt the Tjs between endothelial cells, increasing vascular hyperpermeability [104].

## 4. Mfsd2a Implicated in BBB Permeability during ZIKV Neuroinvasion

Mfsd2a, a member of the major facilitator superfamily (MFS), represents one of the most extensive and diverse membrane transporters [105]. This transporter functions as a versatile facilitator, acting as a uniporter, symporter, and antiporter, responsible for the transport of various essential molecules such as drugs, mono/oligosaccharides, amino acids, and vitamins across cell membranes (Figure 1) [106]. The MFS transporter Mfsd2a, in addition to its counterparts such as Mfsd1, Mfsd2b, and Mfsd3, consists of 12 transmembrane helices (TMs) organized into two pseudosymmetric six-helix bundles known as the N-terminal domain (TMs 1–6) and the C-terminal domain (TMs 7–12) [107]. The predominant mechanism among MFS transporters, including Mfsd2a, involves a “rocker-switch” mechanism that facilitates the import/export of water-soluble molecules [108]. This mechanism involves rigid body movements of the N- and C-terminal domains around a central substrate binding site, exposing them alternately to the membrane’s extracellular (EC) and intracellular (IC) sides. These alternating states, termed outward-facing and inward-facing states, respectively (OFS and IFS), are linked within the transport cycle by an occluded state (OcS) in which the substrate binding site remains inaccessible from either side of the membrane [109].

Endothelial cells that form the BBB and BRB have a strikingly high concentration of Mfsd2a [110]. This integral membrane transporter plays a central role in facilitating Na+-dependent uptake of omega-3 polyunsaturated fatty acid (ω−3 fatty acids). It prevents the formation of caveolae vesicles, which are essential for the maintenance of membrane lipid composition in CNS endothelial cells, particularly docosahexaenoic acid (DHA), in the form of zwitterionic lysolipids such as lysophosphatidylcholine (LPC-DHA) [111]. DHA is not only the predominant ω−3 fatty acid in brain tissue, which protects the integrity of the BBB, but is also an essential component of cell membranes, structurally and functionally interacting with cholesterol and integral membrane proteins that are critical for brain growth, learning, and vision [112,113]. Its high accumulation in the CNS, particularly at neuronal synapses, underlines its indispensability [113]. Insufficient DHA levels correlate with various CNS disorders, including learning and memory disorders, dyspraxia, and dyslexia [114]. In human studies, a deficiency of Msfd2a led to a decrease in the amount of DHA in the brain, resulting in increased transcytosis in several severe neurological disorders, including neonatal neuronal cell death, cognitive impairment, microcephaly [115,116], Alzheimer’s disease, and intracranial hemorrhage [106]. These abnormalities are due to BBB leakage from ZIKV. A recent study in mice has revealed the unique requirement of Mfsd2a for average brain growth and DHA accumulation at the postnatal BBB, as demonstrated by the targeted deletion of Mfsd2a specifically in vascular endothelial cells and the inducible vascular endothelial-specific deletion of Mfsd2a [117]. 

According to recent research, Mfsd2a may moderate the transcytosis of ZIKV. For instance, ZIKV reduced Mfsd2a levels, impairing brain development and DHA supplementation in hBMECs and the developing mouse brain [118]. By genetically ablating Mfsd2a, a leaky BBB is observed from the embryonic stage to maturation while maintaining the usual vascular network pattern, as it is only expressed on CNS endothelial cells [119]. Recently, Jia Zhou et al. [118] demonstrated that ZIKV infection or overexpression of the viral E protein inhibits the uptake of lysophosphatidylcholine (LPC) into cells via Mfsd2a. Lipidomic research also showed that ZIKV infection is associated with reduced DHA-related lipid levels. DHA supplementation can partially reverse ZIKV-induced growth restriction and microcephaly by positively influencing the expression of the transport receptor Mfsd2a. In addition, another study showed that, during ZIKV infection, the protein level of Mfsd2a, rather than the mRNA amount, was perturbed, and the downregulation of Mfsd2a protein responded somewhat to ZIKV. This regard indicated a clear molecular link between ZIKV infection and nervous system disease via Mfsd2a.

An additional study revealed that the inhibition of the ω-3 fatty acid transporter Mfsd2a can facilitate drug delivery across the BBB to treat various diseases such as brain tumors, stroke, and Alzheimer’s disease [118,120]. An electron microscopic analysis of Mfsd2a/mice without visible TJ abnormalities showed a significant increase in the vesicular transcytosis of endothelial cells in the CNS [119]. Therefore, the brain relies on plasma-derived DHA obtained by transport across the BBB [121]. Interestingly, while research has traditionally focused on the role of TJs in BBB function [122], more recent studies have revealed the selective transport of DHA by endothelial cells as the primary route of transport to the brain [121]. However, the reliable and efficient way to control the ZIKV and the BBB regarding this potential transcytosis by regulating Mfsd2a remains unclear.

## 5. Endothelial Cells Use Transcytosis as an Alternative Transport System at BBB

Active and passive endothelial cells facilitate the proper flow of nutrients and regulatory chemicals into the brain. Under typical circumstances, solutes can passively flow via tiny intercellular pores in the TJs. Recent research suggests that claudins are the molecules that affect this transport and generate the pore-forming structures in the TJs of the BBB [123]. Transcellular transport offers more opportunities for drug delivery than this pathway, as early-stage CNS diseases have no BBB abnormalities. In polarized cells, small molecule transport across the cell is standard. Hydrophobic compounds with a molecular weight of less than 500 Da can diffuse transcellularly from the systemic circulation into the brain parenchyma after bypassing the multidrug resistance efflux pumps of the P-gp type [124]. However, specific transporters are required to transport nutrients. For instance, glucose is transported by the glucose transporter (GLUT1) [124], while amino acids, nucleosides, and some drugs are transported by large neutral amino acid transporters (LAT1) [124,125]. The precise mechanism behind this process is not yet fully understood.

Transcytosis is a transcellular pathway that transports molecules using vesicles found in many different cell types, from neurons to intestinal cells to osteoclasts and endothelial cells [126,127]. The macromolecules are not only endocytosed by the vesicles on one side of the cell or transported in vesicles but also exocytosed and released on the other side of the cell [126]. A recent study has shown that there are two types of transcytosis in CNS endothelial cells: (1) receptor-mediated transcytosis (RMT), which is mediated by the binding of ligands to receptors and mediates endocytosis in the case of insulin and transferrin; and (2) adsorptive-mediated transcytosis (AMT), which is facilitated by charged interactions between the molecules and the plasma membrane such as albumin [127]. Both clathrin-mediated, a process ubiquitous in all cell types involved in the endocytosis of cargo by clathrin-coated pits, and caveolae-mediated endocytic mechanisms are essential at the BBB. Clathrin-mediated endocytosis is the primary way that flaviviruses enter human host cells, including ZIKV and DENV. A modification follows this process in the envelope shape, membrane fusion, and the release of viral DNA [128]. In agreement with what is known about sorting systems in other cells, recent research has shown that the BBB may also possess these mechanisms [129].

The efflux transport mechanisms of brain capillary endothelial cells, which remove unwanted chemicals from the brain and pass them into the bloodstream, support the barrier properties of the BBB [126]. Drug efflux from the brain has been associated with molecules resistant to many drugs, monocarboxylate transporters, and organic anion transporters/organic anion transport polypeptides [1]. The efficacy of drugs targeting the CNS is thus limited by the activity of these efflux transporters [126,130]. Adenosine triphosphate-binding cassette (ABC) and solute carrier (SLC) transporters are the two main types of drug transporters that make up most of the drug transporter population. The best-studied BBB transporter of the ABC family is the P-glycoprotein (P-gp), an active transporter that couples efflux to ATP hydrolysis via concentration gradients [131]. The multidrug resistance gene 1 (*MDR1*) encodes P-gp, and intracellular variables and environmental toxins control its activity. The structural and functional tightness of the BBB must be overcome to allow for effective drug transport into the brain. Both paracellular and transcellular pathways are used to overcome the BBB [126].

In the CNS, pericytes are critical for BBB integrity, blood vessel formation, and the regulation of immune cell entry [132]. The basement membranes of endothelial cells are embedded in pericytes, enabling extensive signaling between the two cell types. It also surrounds the endothelial cells of the CNS. Studies have shown that the integrity of the CNS endothelial cell barrier depends on a precise ratio of pericytes to endothelial cells. Both pericyte-deficient mice, which are generated by altering the signaling pathway that generally attracts pericytes [133], and the pericyte-specific transcription factor Foxf2, which increases pericyte density [134], exhibit leaky barriers and attribute the leakiness to an increase in transcytosis [134]. These studies suggest that pericytes are essential elements of the neurovascular unit that regulate transcytosis in CNS endothelial cells to provide barrier properties. However, it is unclear which specific genes within pericytes and which paracrine signaling pathways between pericytes and endothelial cells regulate barrier properties. Future research will help define the processes that regulate transcytosis at the BBB by discovering specific receptor–ligand interactions between these cells and genes downstream of the transcription factor Foxf2, which regulates transcytosis in CNS endothelial cells.

In addition, ZIKV infection can infect and release the apical and basolateral surfaces of BMECs [135]. Several in vitro experiments show that ZIKV can replicate and cross the BBB without affecting its permeability and integrity. The lower chamber of the trans-well system contained ZIKV RNA, although the integrity of the endothelium was preserved. These studies demonstrate that ZIKV can penetrate or pass through the BBB via transcytosis, paracellular, or basolateral viral release. Positively charged ZIKV particles are attached to negatively charged brain endothelial cell membranes by a charge-based process, leading to transcytosis. The permeability of MECs was not affected by the release of type I and type III IFNs and inflammatory cytokines, but transinfection was facilitated. These results suggest that the extravasation of ZIKV through the MEC monolayer can be effectively stopped by drugs that inhibit ZIKV replication or transcytosis.

## 6. Conclusions

This review explores the intricate mechanisms by which ZIKV invades the brain through the BBB by crossing endothelial cells. The endothelial cell transcytosis pathways serve as an alternative transport system and demonstrate perspectives into the penetration mechanisms of ZIKV and potential therapeutic targets. This work also explains the complex interplay between the ZIKV infection and host defense mechanisms and highlights the importance of vascular permeability in pathogenesis. By elucidating the interplay between ZIKV and endothelial cells, we revealed the crucial role of pericytes in regulating transcytosis and the importance of mfsd2a in vascular permeability. To date, developing a vaccine or proper treatment to cure ZIKV infection is still challenging, and there have been no effective methods other than medications to prevent viral replication and neural abnormalities. Our findings not only improve our understanding of the neuropathogenesis of ZIKV but also offer promising avenues for developing targeted interventions to mitigate ZIKV-associated neurological complications. Overall, this review underscores the importance of understanding the complexities of ZIKV neuroinvasion to develop effective disease management and prevention strategies.

## Figures and Tables

**Figure 1 viruses-16-00629-f001:**
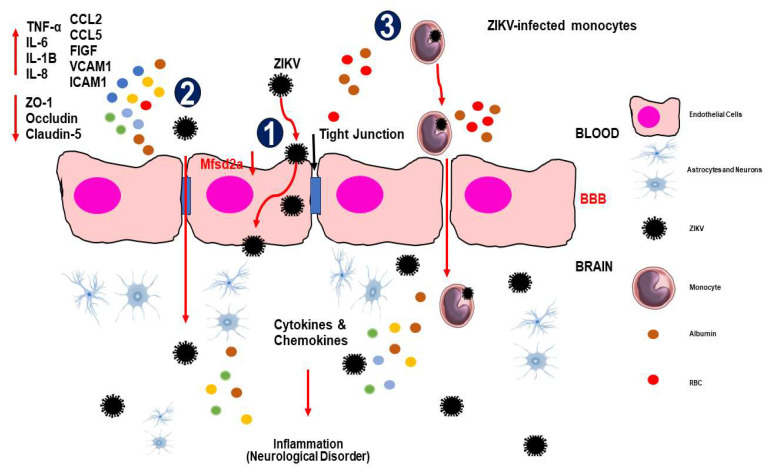
The mechanisms of ZIKV invade the central nervous system (CNS). (1) Transcellular transport via infection or transcytosis facilitated by ZIKV-induced Mfsd2a breakdown within BBB endothelial cells. (2) ZIKV is paracellularly trafficked across the BBB by upregulating pro-inflammatory cytokines, chemokines, adhesion molecules, and growth factors and downregulating tight junction proteins, which modifies the integrity and permeability of the endothelial barrier. (3) The Trojan horse method allows ZIKV-infected monocytes to pass through the BBB. When ZIKV enters the CNS, it infects brain tissue, including astrocytes and microglial cells, which release cytokines and chemokines that cause inflammation/injury. Abbreviations: TNF-α, tumor necrosis factor alpha; IL, interleukin; CCL, chemokine (C-C motif) ligand; FIGF, C-Fos-induced growth factor (vascular endothelial growth factor D); VCAM, vascular cell adhesion molecule; ICAM, intercellular adhesion molecule.

**Figure 2 viruses-16-00629-f002:**
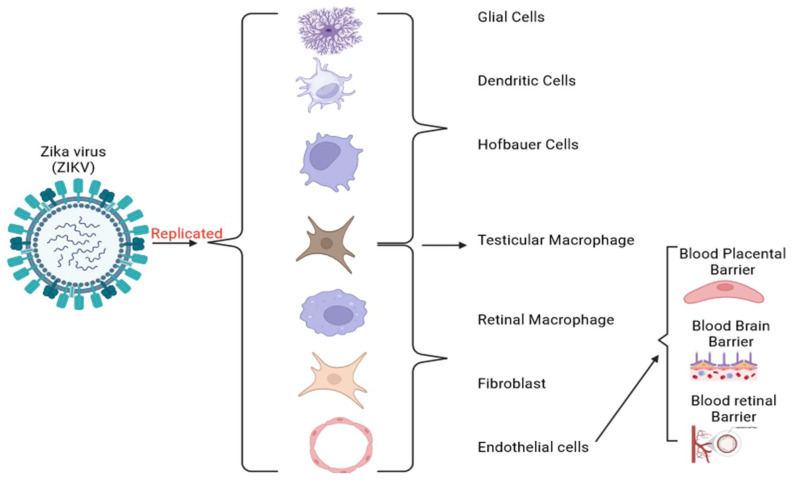
Replication of the ZIKV and its target human cells. ZIKV infects pregnant women by targeting trophoblasts and Hofbauer cells. It can cross the BPB and cause an immunologic response and brain cell death, impairing neurogenesis and leading to microcephaly [64]. In brain development, ZIKV infects radial glial cells and intermediate progenitor cells constituting the brain and CNS [65]. ZIKV also overcomes the BBB by infecting the brain endothelial cells and altering the tight junction proteins.

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
