# Peer review of "Exploring Zika Virus Impact on Endothelial Permeability: Insights into Transcytosis Mechanisms and Vascular Leakage"

_viruses, 2024, doi:10.3390/v16040629_

Round 1

Reviewer 1 Report

Comments and Suggestions for Authors

Journal Viruses (ISSN 1999-4915)

Manuscript ID: viruses-2925879

Type: Review

Title: Exploring Zika virus impact on endothelial permeability: insights into transcytosis mechanisms and Vascular Leakage

Authors: Dama Faniriantsoa Henrio Marcellin , Heng Li , Huang Jufang 

(1)This is a great review that is well-written.  The authors have included many references and the most up-to-date studies and findings but are missing some that should be included:

·       Langerak T, Broekhuizen M, Unger PA, Tan L, Koopmans M, van Gorp E, Danser AHJ, Rockx B. Transplacental Zika virus transmission in ex vivo perfused human placentas. PLoS Negl Trop Dis. 2022 Apr 20;16(4):e0010359. doi: 10.1371/journal.pntd.0010359. PMID: 35442976; PMCID: PMC9060339.

·       Zanluca C, de Noronha L, Duarte Dos Santos CN. Maternal-fetal transmission of the zika virus: An intriguing interplay. Tissue Barriers. 2018 Jan 2;6(1):e1402143. doi: 10.1080/21688370.2017.1402143. Epub 2018 Jan 25. PMID: 29370577; PMCID: PMC5823548.

·       Tan LY, Komarasamy TV, James W, Balasubramaniam VRMT. Host Molecules Regulating Neural Invasion of Zika Virus and Drug Repurposing Strategy. Front Microbiol. 2022 Mar 4;13:743147. doi: 10.3389/fmicb.2022.743147. PMID: 35308394; PMCID: PMC8931420.

·       Todorovski T, Mendonça DA, Fernandes-Siqueira LO, Cruz-Oliveira C, Guida G, Valle J, Cavaco M, Limas FIV, Neves V, Cadima-Couto Í, Defaus S, Veiga AS, Da Poian AT, Castanho MARB, Andreu D. Targeting Zika Virus with New Brain- and Placenta-Crossing Peptide-Porphyrin Conjugates. Pharmaceutics. 2022 Mar 29;14(4):738. doi: 10.3390/pharmaceutics14040738. PMID: 35456572; PMCID: PMC9032516.

·       Xu Y, He Y, Momben-Abolfath S, Eller N, Norton M, Zhang P, Scott D, Struble EB. Entry and Disposition of Zika Virus Immune Complexes in a Tissue Culture Model of the Maternal-Fetal Interface. Vaccines (Basel). 2021 Feb 11;9(2):145. doi: 10.3390/vaccines9020145. PMID: 33670199; PMCID: PMC7916977.

(2)        Page 3: Add space in line 113 before reference [22].

(3)        Page 8: Add space line 352 before reference [96].

(4)        Figures are quite nice and add a lot to the article!  There are many complex pathways involved in Zika Virus pathogenesis and the figures are very helpful.  However, the color numbers should be different, hard to see the white color in gray.

(5)        As a reader, I like to see abbreviations defined in the legend, so the figure can be read without going back and forth to main text.   Some abbreviations are defined in the figures, but not all.

Reviewer 2 Report

Comments and Suggestions for Authors

The review presented by Marcellin and colleagues wants to give insights on ZIKA Virus infection dependent endothelial damage, as the title suggests. Nevertheless, only 3 out of 6 sections really focus on the subject.

Section 2 is too long and information on zikv symptoms should be included in the introduction, section 3, entitled "immunopathogenesis of ZIKV" describes  the innate response to the virus and suggests  a link between astrocytes' infection and lymphocyte infiltration. The message of section 3 is not clear.

section 4, 5, and 6 only marginally describe  ZIKV impact on the highlighted mechanisms

I suggest to shorten the first part of the reviev (section 1, 2, and 3 to be included in the introduction), and to go deeper in the literature for what concern the interaction of ZIKV with the endothelial cells and biological barriers.

Comments on the Quality of English Language

English revision is needed 
